# Azithromycin Augments Bacterial Uptake and Anti-Inflammatory Macrophage Polarization in Cystic Fibrosis

**DOI:** 10.3390/cells13020166

**Published:** 2024-01-16

**Authors:** Abdullah A. Tarique, Neeraj Tuladhar, Dean Kelk, Nelufa Begum, Richard M. Lucas, Lin Luo, Jennifer L. Stow, Claire E. Wainwright, Scott C. Bell, Peter D. Sly, Emmanuelle Fantino

**Affiliations:** 1Child Health Research Centre (CHRC), The University of Queensland, Brisbane, QLD 4101, Australiap.sly@uq.edu.au (P.D.S.); e.fantino@uq.edu.au (E.F.); 2Institute for Molecular Bioscience (IMB), The University of Queensland, Brisbane, QLD 4067, Australia; 3Respiratory and Sleep Medicine, Queensland Children’s Hospital, Brisbane, QLD 4101, Australia; 4Thoracic Medicine, The Prince Charles Hospital, Brisbane, QLD 4032, Australia

**Keywords:** cystic fibrosis, CFTR, phagocytosis, bacterial killing, *Pseudomonas aeruginosa*, macrophage polarization, ERK1/2, azithromycin, antibiotic resistance

## Abstract

Background: Azithromycin (AZM) is widely being used for treating patients with cystic fibrosis (pwCF) following clinical trials demonstrating improved lung function and fewer incidents of pulmonary exacerba-tions. While the precise mechanisms remain elusive, immunomodulatory actions are thought to be involved. We previously reported impaired phagocytosis and defective anti-inflammatory M2 macrophage polarization in CF. This study systematically analyzed the effect of AZM on the functions of unpolarized and M1/M2 polarized macrophages in CF. Methods: Monocytes, isolated from the venous blood of patients with CF (pwCF) and healthy controls (HCs), were differentiated into monocyte-derived macrophages (MDMs) and subsequently infected with *P. aeruginosa*. *P. aeruginosa* uptake and killing by MDMs in the presence or absence of AZM was studied. M1 and M2 macrophage polarizations were induced and their functions and cytokine release were analyzed. Results: Following AZM treatment, both HC and CF MDMs exhibited a significant increase in *P. aeruginosa* uptake and killing, however, lysosomal acidification remained unchanged. AZM treatment led to higher activation of ERK1/2 in both HC and CF MDMs. Pharmacological inhibition of ERK1/2 using U0126 significantly reduced *P. aeruginosa* uptake in HC MDMs. M1 macrophage polarization remained unaffected; however, AZM treatment led to increased IL-6 and IL-10 release in both HC and CF M1 macrophages. AZM also significantly increased the phagocytic index for both pHrodo *E. coli* and *S. aureus* in CF M1 macrophages. In CF, AZM treatment promoted anti-inflammatory M2 macrophage polarization, with an increased percentage of CD209^+^ M2 macrophages, induction of the M2 gene *CCL18*, along with its secretion in the culture supernatant. However, AZM d’d not restore endocytosis in CF, another essential feature of M2 macrophages. Conclusions: This study highlights the cellular functions and molecular targets of AZM which may involve an improved uptake of both Gram-positive and Gram-negative bacteria, restored anti-inflammatory macrophage polarization in CF. This may in turn shape the reduced lung inflammation observed in clinical trials. In addition, we confirmed the role of ERK1/2 activation for bacterial uptake.

## 1. Introduction

Chronic pulmonary infection and excessive inflammation are the leading causes of progressive pulmonary damage in cystic fibrosis (CF) lung disease. Managing chronic infection and limiting the excessive production of inflammatory mediators are the major therapeutic strategies for slowing the deterioration of lung function and improving survival in CF. Randomized controlled trials with azithromycin (AZM) showed an improvement in lung function, fewer pulmonary exacerbations and hospital stays, reduced *Pseudomonas aeruginosa* colonization rates, and less intravenous antibiotic usage in patients with CF [1,2,3]. However cellular and mechanistic evidence regarding the mode of action of AZM in CF was scarce. These improvements most likely came from the immunomodulatory properties of macrolides [4]. AZM significantly reduced pro-inflammatory cytokine release (IL-1β, CCL2, and TNFα) and enhanced the polarization of anti-inflammatory M2 macrophages in CF mice homozygous for the F508del mutation as compared with macrophages from wild-type mice [5]. AZM has also been shown to improve phagocytosis ex vivo in macrophages obtained from patients with chronic obstructive pulmonary disease (COPD) [6]. However, the cellular mechanisms by which AZM exhibited immunomodulatory effects in CF remain largely unexplored.

Macrophages represent the first line of immune defense against invading pathogens in the lungs and are divided into lung-resident alveolar macrophages (AMs) and recruited monocyte-derived macrophages (MDMs). CF sputum analysis showed a shift from AMs (0.4%) to MDMs (19%) [7]. When recruited to the lungs to deal with invading pathogens, MDMs play crucial roles in the initiation and resolution of pulmonary inflammation via their pro-(M1) or anti-(M2) inflammatory phenotypes, respectively [8]. In the early stage of an inflammatory response, the M1 macrophage phenotype, activated via the classical pathway, uptakes the bacteria, initiates inflammation, releases pro-inflammatory mediators, and kills bacteria by phagocytosis [9]. Later, inflammation-resolving M2 macrophages release anti-inflammatory cytokines and clear apoptotic cells by endocytosis to return to homeostasis [9]. A dynamic equilibrium between M1 and M2 macrophages is crucial to maintain tissue homeostasis in the lungs. We have previously shown defective phagocytosis by M1 macrophages and impaired polarization to anti-inflammatory (M2) phenotypes in monocyte-derived macrophages (MDMs) obtained from patients with CF (pwCF) [10].

The present study was undertaken to investigate whether defective macrophage function and polarization in CF were improved by AZM. We differentiated peripheral blood monocytes into MDMs, polarized them into the M1 phenotype, and studied macrophage functions, including *P. aeruginosa* uptake and killing ability as *P. aeruginosa* is the leading organism found in the lungs of pwCF. We also studied the relationship between bacterial uptake and ERK1/2 activation, and whether AZM modulates ERK1/2 activation and thereby bacterial uptake.

The primary objective of this study was to assess whether AZM could ameliorate the defective macrophage function and polarization observed in CF. We therefore investigated the effect of AZM on the functions of unpolarized MDMs, LPS-induced M1 and IL-13-induced M2 macrophages. Bacterial killing ability was assessed against *E. coli*, *P. aeruginosa*, and *S. aureus*, where the latter two are also known to be the predominant bacteria in CF-affected lungs. Additionally, this study explored ERK1/2 as the key molecular pathway required for bacterial killing and clearance.

## 2. Methods

### 2.1. Study Participants

Buffy coats from healthy controls (HCs) (n = 15) aged 20–40 years were obtained from the Australian Red Cross Blood Service. 17 adults and children with CF, carrying at least one F508del allele were recruited from the CF clinics at The Prince Charles Hospital (TPCH) and Queensland Children’s Hospital (QCH), Brisbane, Australia, respectively (Table 1).

### 2.2. Ex Vivo Macrophage Differentiation and Polarization

Peripheral blood mononuclear cells (PBMCs) were isolated from buffy coats or peripheral blood obtained from pwCF using lymphoprep (AxisShiled, Dundee, UK). Monocytes were enriched by CD14^+^ microbeads following the manufacturer’s instructions (Miltenyi Biotec, Bergisch Gladbach, Germany). Macrophage differentiation and polarization were performed as previously described [9]. Briefly, CD14^+^ monocytes were differentiated into unpolarized macrophages (MDMs) by a 6-day stimulation with granulocyte-macrophage colony-stimulating factor (GM-CSF) (50 ng/mL, Miltenyi, Biotec, Bergisch Gladbach, Germany) in RPMI-1640 plus 10% FBS (ThermoFisher, Waltham, MA, USA) and 1% penicillin-streptomycin-amphotericin B (Lonza, Walkersville, MD, USA). Medium was refreshed on day 3 with GM-CSF. M1 and M2 polarizations were induced by *E. coli* LPS (20 ng/mL, Sigma, Livonia, MI, USA) and IL-13 (20 ng/mL, ThermoFisher, Waltham, MA, USA), respectively, for 2 days. 5 μg/mL of AZM (Sigma, USA) was added to the culture medium during differentiation and before polarization.

### 2.3. AZM Cytotoxicity

AZM effects on the host cells are dose dependent [4]. Therefore, the cytotoxicity of AZM at different concentrations was assessed on MDMs using a membrane-impermeant 7-Aminoactinomycin D (7-AAD) DNA binding dye staining (BD, Franklin Lakes, NJ, USA). The median fluorescence intensity (MFI) of 7-AAD was calculated using FlowJo 8.1.

### 2.4. P. aeruginosa Uptake and Killing

Cells were infected with a highly virulent *P. aeruginosa* strain PA14 at MOI 10 for 1 h at 37 °C in the absence of any antibiotic. After 1 h, the infection medium was removed and RPMI containing gentamycin (200 μg/mL, ThermoFisher, Waltham, MA, USA) was added for 1 h to kill any extracellular bacteria. This 2 h time-point was defined as the bacterial uptake (Figure 1A). Cells were then left in a low-dose gentamycin (20 μg/mL) containing RPMI for next 2 h to assess the killing of ingested bacteria. Cells were then lysed with 1% saponin and plated onto LB agar plate. The killing index was calculated as (CFU at 2 h − CFU at 4 h) × 100/CFU at 2 h.

### 2.5. Lysosome Staining

Cells were infected with PA14 at MOI 10 for 1 h at 37 °C. After 1 h, the infection medium was removed and RPMI containing gentamycin (200 μg/mL) and 500 nM of LysoTracker green (Thermofisher, Waltham, MA, USA) was added for 30 min to kill any extracellular bacteria and to stain the lysosomes. Cells were acquired on BD Fortessa (Franklin Lakes, NJ, USA). The median fluorescence intensity (MFI) of LysoTracker green was calculated using FlowJo.

### 2.6. Western Blot

MDMs stimulated with LPS for 30 min were lysed by RIPA buffer containing protease inhibitor cocktail (Roche, Indianapolis, IN, USA) and phosSTOP phosphatase inhibitors (Roche, Indianapolis, IN, USA). The total protein was quantified using a BCA protein assay (ThermoFisher, Waltham, MA, USA). Immunoblotting was performed on a 10% SDS-PAGE and the proteins were transferred to a PVDF membrane, blocked with 5% (*w*/*v*) skim milk in TBST and incubated overnight at 4 °C with primary antibodies (Table 2), after washing, a secondary antibody was added. The membrane was then developed with an ECL substrate (Bio-Rad, Hercules, CA, USA) and imaged using the ChemiPro Imaging System (Cleaver, Rugby, UK).

### 2.7. ERK1/2 Inhibition

MDMs were pre-treated with 10 or 20 μM of U0126, a potent ERK1/2 inhibitor (Sigma, USA) for at least an hour prior to PA14 infection.

### 2.8. Flow Cytometric Analysis of Cell Surface Receptors and M1/M2 Markers

Fully mature MDMs were analyzed for the surface expression of TLR4, IL-13Rα1, and IL-4Rα, essential for M1 and M2 polarization, respectively. The percentage of CD80^+^ M1 macrophages and that of CD209^+^ M2 macrophages was analyzed using anti-human CD80 PE and anti-human CD209 BV450. 7-AAD staining was performed to test the cell viability [9]. The data were acquired using a BD LSR-Fortessa (Franklin Lakes, NJ, USA). All analyses were performed on Flowjo.

### 2.9. Phagocytosis and Endocytosis

Phagocytosis was studied by incubating M1 macrophages with pHrodo green *E. coli* and pHrodo red *S. aureus* bioparticles (LifeTech, Carlsbad, CA, USA) at 37 °C for 90 min, following the manufacturer’s instructions. Endocytosis was determined by incubating the M2 macrophages with AF647-dextran (10 KD) (LifeTech, Carlsbad, CA, USA) at 37 °C for 90 min, following the manufacturer’s instructions. The cells were then washed and acquired on a BD Fortessa (Franklin Lakes, NJ, USA). The phagocytic or endocytic index was calculated by normalizing the corresponding MFI with either CD80^+^ or CD209^+^ cells, respectively, as previously reported [11].

### 2.10. Cytokine Quantification

M1-specific pro-inflammatory cytokines, such as, IL-6, IL-8, IL-10, and TNFα, were quantified in a culture supernatant using alphaLISA (PerkinElmer, Waltham, MA, USA), as per manufacturer’s instructions.

### 2.11. Statistical Analysis

The Wilcoxon signed-rank test was used to determine the statistical difference between two outcome variables with paired data. For a between-group comparison, the Wilcoxon rank-sum (Mann–Whitney) test was used for two groups, and the Kruskal–Wallis rank test was used for more than two groups. When the overall significance was observed for more than two groups, Dunn’s multiple comparisons (without adjustment) was used to determine the pair-wise significance difference. Statistical significance was determined at the 0.05 level. The data are presented as median (25th–75th percentile) unless stated otherwise. All analyses were performed using Graph-Pad 7 (San Diego, CA, USA). 

## 3. Results

### 3.1. Azithromycin Cytotoxicity

Under light microscopy, MDMs treated with 50 μg/mL of AZM were comparatively smaller in size than the vehicle-treated control MDMs. Flow cytometric analysis showed that MDMs treated with 2 and 5 μg/mL of AZM showed minimal 7-AAD staining. With increasing AZM conc, the 7-AAD staining increased indicating AZM cytotoxicity (Appendix A) Since a mean peak concentration of 3.89 μg/mL has been reported in bronchial mucosa following AZM administration [12], we chose to use 5 μg/mL of AZM for all subsequent experiments.

### 3.2. AZM Increases the Uptake and Killing of P. aeruginosa by CF Macrophages

We previously observed impaired phagocytosis of *E. coli* by LPS-induced CF M1 macrophages [10]. In this study, we used unpolarized MDMs and observed a similar phenomenon. Uptake of *P. aeruginosa* significantly decreased in unpolarized CF MDMs compared to HC MDMs (*p* = 0.001). AZM treatment led to a significant increase in *P. aeruginosa* uptake in both HC (*p* = 0.002) and CF MDMs (*p* = 0.003) (Figure 1B), however the uptake in AZM-treated CF MDMs still remained significantly below the HC MDMs that were not exposed to AZM (*p* = 0.007). The killing index of *P. aeruginosa* was only very modestly enhanced in CF MDMs following AZM treatment (*p* = 0.02), but not in HCs (Figure 1C). As lysosomal acidification, a key mechanism of the MDMs to kill the engulfed bacteria, was previously reported to be defective in CF MDMs by our group and others [10,13], we stained the lysosomes of *P. aeruginosa*-infected MDMs with LysoTracker with the aim of finding whether AZM was able to alter the lysosomal acidification. The median fluorescent intensity (MFI) of the LysoTracker slightly increased, though not significantly, in both HC and CF MDMs (Appendix A). Altogether, this study confirms the role of AZM in *P. aeruginosa* uptake, but not in killing.

### 3.3. ERK1/2 Activation Is Pivotal for P. aeruginosa Uptake

The activation of extracellular-signal-regulated kinase (ERK1/2) has been reported for efficient bacterial phagocytosis [14,15]. ERK1/2 activation was also observed in *P. aeruginosa*-infected CF epithelial cells [16]. We therefore analyzed for a direct link between ERK1/2 activation and *P. aeruginosa* uptake in HC and CF MDMs. Infection with *P. aeruginosa* led to a significant activation of ERK1/2 in both HC (*p* = 0.03) and CF (*p* = 0.03) MDMs (Figure 2A). The AZM treatment increased ERK1/2 activation even more in both HC and CF (*p* = 0.03) MDMs. To confirm the role of ERK1/2 in *P. aeruginosa* uptake, HC MDMs were treated with U0126, an ERK1/2 inhibitor, 1 h prior to *P. aeruginosa* infection. U0126 treatment led to a significant reduction in *P. aeruginosa* uptake in HC MDMs in a dose-dose-dependent manner (Figure 2B), confirming the role of ERK1/2 activation in bacterial uptake.

### 3.4. AZM Neither Reduced Pro-Inflammatory (M1) Macrophage Polarization nor Pro-Inflammatory Cytokine Secretion

We then investigated the effect of AZM in LSP-stimulated M1 macrophage polarization and its cytokine release. M1 macrophage polarization was not affected by AZM in either HC or CF (Figure 3A). IL-6 and IL-10 release significantly increased in both HC and CF M1 macrophages following AZM treatment; however, IL-8 and TNF-α levels remained unchanged (Figure 3B–E, IL-8, TNF-α data not shown). LPS-induced M1 macrophages are a more effective phagocytic than unpolarized MDMs [9]. We then tested the phagocytic ability of these AZM-treated M1 macrophages using pHrodo-labelled Gram-negative *E. coli* and Gram-positive *S. aureus*. The phagocytic index for both *E. coli* and *S. aureus* was significantly lower in CF M1 macrophages compared to HC M1 macrophages (Figure 3D,E). This observation aligns with the data obtained from the PA14 infection data shown in Figure 1B. AZM treatment led to an enhanced phagocytic index for both bacteria in the CF M1 macrophages (Figure 3D,E). These data suggest that AZM enhances bacterial uptake irrespective of the Gram-positive and Gram-negative bacterial strains. To understand the molecular mechanism of the AZM treatment in M1 macrophages, we analyzed the activation of NFκB. The phosphorylation of NFκB (p65) was unaffected by AZM (Appendix A).

### 3.5. AZM Promotes Anti-Inflammatory (M2) Macrophage Polarization in CF

Macrophages play a critical role in the resolution of inflammation and restoration of homeostasis through their anti-inflammatory (M2) attributes. We previously reported that M2 polarization was deficient in CF [10]. In this study, we analyzed the effect of an AZM treatment on CF macrophages’ M2 polarization by measuring the level of surface marker expression, the release of *CCL18* (an M2-signature gene that we previously reported on [9]), and endocytosis, another M2-macrophage specific function which is associated with the removal of apoptotic cells. AZM treatment resulted in an increased percentage of CD209^+^ M2 macrophages in CF macrophages (Figure 4A,B), and an enhanced release of CCL18 in both HC and CF M2 macrophages (Figure 4C). However, endocytosis, another M2-macrophage specific function which is associated with removal of apoptotic cells, remained unaffected (Figure 4D). Thus, our data suggest that AZM partially restores anti-inflammatory (M2) macrophage function in CF.

## 4. Discussion

Failure to clear bacteria from the lungs and exaggerated inflammation are common in CF [17]. Although RCTs trials with AZM showed an improvement in lung function and reduced *Pseudomonas aeruginosa* colonization rates [1,2,3], cellular and mechanistic evidence regarding the mode of action of AZM in CF is scarce. We herein demonstrate that AZM significantly augments the uptake of *P. aeruginosa* by both HC and CF MDMs, and the killing index significantly increased for CF MDMs too. However, lysosomal staining, an indication of lysosomal pH, remained unchanged. Mechanistically, AZM-enhanced uptake was associated with ERK1/2 pathway upregulation. An increased phagocytosis of *E. coli* and *S. aureus* was also observed in CF pro-inflammatory M1 macrophages. In addition, AZM partially restored the anti-inflammatory M2 macrophage polarization in CF. Such an enhanced uptake of both Gram-positive and Gram-negative bacteria, which confers an improved bacterial killing in CF and the restoration of anti-inflammatory M2 polarization, may contribute to the clinical improvement seen in pwCF treated with AZM.

Macrophages detect pathogens using pathogen recognition receptors (PRRs), which activate signalling pathways to internalize the pathogen into phagosomes. Phagosomes later fuse with acidic organelle lysosomes which contain reactive oxygen species (ROS), reactive nitrogen species (RNS), proteases, and antimicrobial peptides to facilitate the killing of engulfed pathogens. Elegant studies have demonstrated that alveolar macrophages (AMs) do not proliferate and disappear during infection [18]. Instead, monocyte-derived macrophages (MDMs) are recruited to the lungs to initiate inflammation and are then involved in bacterial killing and inflammation inception [19]. MDMs were found higher in numbers in CF-affected lungs compared with alveolar macrophages [7], confirming the relevance of their use as macrophage models of CF pathogenesis. Using MDMs, we have previously demonstrated a CFTR-dependent defect during phagocytosis in CF macrophages [10]. These functional deficiencies could contribute to an exaggerated pulmonary inflammation by failing to kill and clear bacteria in the CF-affected lungs.

Using unpolarized MDM, we initially analyzed the uptake and killing of *P. aeruginosa* in HC and CF MDMs in the presence or absence of AZM. Similar to our previous study with *E. coli* [10], we herein showed a reduced uptake of *P. aeruginosa* by CF MDMs. The AZM treatment significantly enhanced *P. aeruginosa* uptake in both HC and CF; however, the uptake in AZM-treated CF MDMs still remained significantly lower than in HC MDMs that were not exposed to AZM. Phagolysosomal acidification, the key killing machineries of the MDMs, is tightly regulated by the CFTR channel function, which was previously found to be lower in CF macrophages [13]. In this study, lysosome staining using LysoTracker did not show any substantial increase in AZM-treated HC and CF MDMs compared to the controls. This implies that AZM has no effect on the CFTR channel function. The increased killing index by CF MDMs may be due to a higher uptake of *P. aeruginosa*. Thus, we herein conclude that AZM enhances the bacterial uptake in macrophages but has no role in enhancing the phagolysosomal killing.

Bacterial uptake and phagolysosomal bacterial killing are complicated processes, with the involvement of different signalling pathways. Activation of extracellular-signal-regulated kinase (ERK1/2) has been reported for efficient phagocytosis [14,15]. The inhibition of the ERK1/2 pathway with U0126 resulted in a reduced phagocytosis, confirming the role of ERK1/2 in phagocytosis [20]. In addition, the ERK1/2 pathway is a known target of AZM [21]. Similar to CF epithelial cells [16], ERK1/2 activation was observed after infection in both HC and CF MDMs, and was enhanced following an AZM treatment. ERK1/2 inhibition using U0126 diminished the *P. aeruginosa* uptake in HC MDMs in a dose-dependent manner, indicating a direct link to ERK1/2 activation in *P. aeruginosa* uptake.

The M1 pro-inflammatory MDM phenotype is associated with an increased inflammatory response. Data from the present study on the effects of AZM on M1 polarization differ from a previous study conducted in a murine model of CF that reported reduced M1 polarization in peritoneal macrophages using NO production as a marker of polarization [5]. However, we observed no reduction in the percentages of CD80^+^ M1 macrophages in both HC and CF M1 macrophages following an AZM treatment. NFκB is a key molecular regulator of inflammation and induction of NFκB-dependent effector genes. We did not see any change in NFκB phosphorylation. The inhibition of NFκB phosphorylation was previously observed in human monocytes at a high dose of AZM [22], but we observed that such higher doses of AZM are highly cytotoxic to MDMs. Using the dose of AZM that we used in this study, Haydar et al. observed a significant inhibition of the translocation of NFκB in bone-marrow-derived macrophages [23]. However, Haydar et al. did not analyze cytokine release, whereas we observed a substantial increase in the IL-6 and IL-10 release in HC and CF M1 macrophages following an AZM treatment. These data are consistent with previous studies using macrophage cell lines, showing that ERKs1/2 activation positively regulates IL-10 production [24]. Since CF M1 macrophages exhibited enhanced phagocytosis of pHrodo *E. coli* and *S. aureus* bioparticles, it is important to address whether there is any role that IL-6 or IL-10 play in bacterial phagocytosis. IL-6 has previously been reported to play a role in phagocytosis [25]. Recently Akoumianaki et al. demonstrated the requirement for IL-6 in the trafficking of ERK1/2 to phagosomes [26]. Although other studies with epithelial cells reported a reduced TNF-α and IL-8 release following AZM treatment, this was not observed in our study. Together, our data suggest that AZM induces an IL-6 and IL-10 release, which in turn promotes bacterial phagocytosis by CF M1 macrophages.

We previously reported an impaired polarization of anti-inflammatory (M2) macrophages in CF [10]. Haydar et al. demonstrated that AZM was able to induce M2 macrophage polarization by inhibiting the STAT1 and NFκB signalling pathways [23]. Meyer et al. showed a shift from pro-inflammatory M1 toward anti-inflammatory M2 macrophage polarization by AZM treatment in F508del-CF mice [5]. In agreement with these studies, we observed an increase in M2 polarization in CF following an AZM treatment. CCL18 release was enhanced in both HC and CF M2 macrophages. CCL18 is known to recruit monocytes/macrophages and regulatory T cells to the site of inflammation to maintain homeostasis. Pechkovsky et al. showed that IL-10 enhanced the CCL18 expression in M2 macrophages [27]. In physiological conditions, it is possible that M1 macrophages clear the invading pathogen and release IL-10, which in turn induces M2 polarization. However, we did not find an increase in endocytosis in AZM-treated CF M2 macrophages. AZM has previously been reported to selectively inhibit endocytosis by inhibiting the endocytic uptake and the transport of solutes along the endocytic pathway in the murine macrophage cell line J774 [28]. This may explain why we did not observe any enhancement of endocytosis in CF M2 macrophages. Further research is required to identify the molecular mechanism of enhanced M2 macrophage polarization following an AZM treatment.

A key question raised by our observation of enhanced bacterial uptake and improved M2 macrophage polarization was whether AZM induced CFTR protein expression and chloride channel activity or whether other mechanisms were involved. AZM has previously been reported to increase chloride efflux in CF epithelial cells without increasing CFTR protein or mRNA expression [29]. In our observation, neither CFTR mRNA nor CFTR protein expression, as assessed by flow cytometry, increased in AZM-treated MDMs when compared to the control groups (data not shown). Therefore, the detailed molecular mechanism underlying the improved CF macrophage function and polarization still remains unclear and warrants further studies.

We observed some limitations in our study. While MDMs are appropriate cells to study [30], we assess them remote from their usual site of action. In addition, due to the limited volume of blood that was collected from the patients with CF, we did not have sufficient cells to perform all of the studies on cells from all of the patients. In summary, we provide evidence that the anti-inflammatory effects of AZM may be contributed to via an increase in the bacterial uptake by CF MDMs, increasing the proportion of CF MDMs able to polarize into the inflammation-resolving M2 phenotype and increasing the secretion of the anti-inflammatory cytokines IL-10 and CCL18. Mechanistically, the increased bacterial uptake was mediated by activating the ERK1/2 pathway.

## Figures and Tables

**Figure 1 cells-13-00166-f001:**
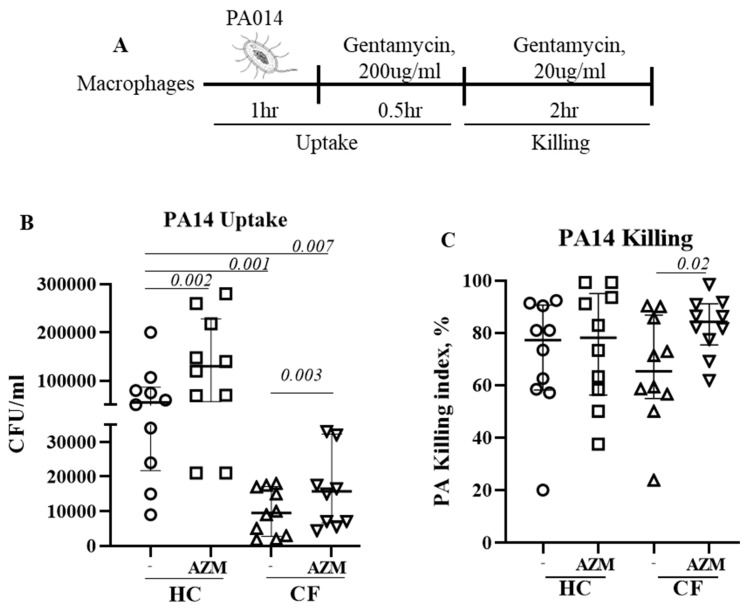
Azithromycin augments *P. aeruginosa* uptake and killing in CF. (**A**) Schematic presentation of PA14 infection in MDMs. Cells were pre-treated with AZM or DMSO and then infected with PA14 at MOI 10 for 1 h in the absence of AZM. After one hour, the infection medium was removed and RPMI containing gentamycin (200 μg/mL) was added for 30 min to kill any extracellular bacteria. Cells were either immediately lysed or left for 2 h in a low-dose gentamycin (20 μg/mL) containing media for 2 h, then lysed with saponin, serially diluted, and then plated onto LB agar plates. Colony counts were conducted for PA14 uptake (**B**) and killing (**C**). The killing index was measured as described in the methods section. Each dot represents an individual healthy donor (HC = 10) or pwCF (n = 9). Wilcoxon signed-rank test and Mann–Whitney tests were performed. The data were shown as the median (25%, 75%).

**Figure 2 cells-13-00166-f002:**
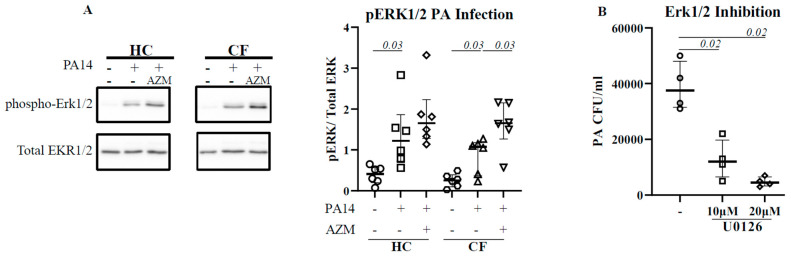
AZM augments ERK1/2 activation in CF. Macrophages (M0) were differentiated in the presence or absence of AZM and infected with PA14 for 30 min at MOI 10. The cells were lysed. The phosphorylation of ERK1/2 was analyzed by western blot (**A**). Phosphorylation was normalized by the level of uninfected macrophages. To confirm the role of ERK1/2 in phagocytosis, MDMs were pre-treated with U0126, followed by PA14 infection as mentioned in the methods (**B**). Each symbol represents an individual healthy donor (HC = 6) or pwCF (n = 6). A Wilcoxon signed-rank test was performed. The data are shown as the median (25%, 75%).

**Figure 3 cells-13-00166-f003:**
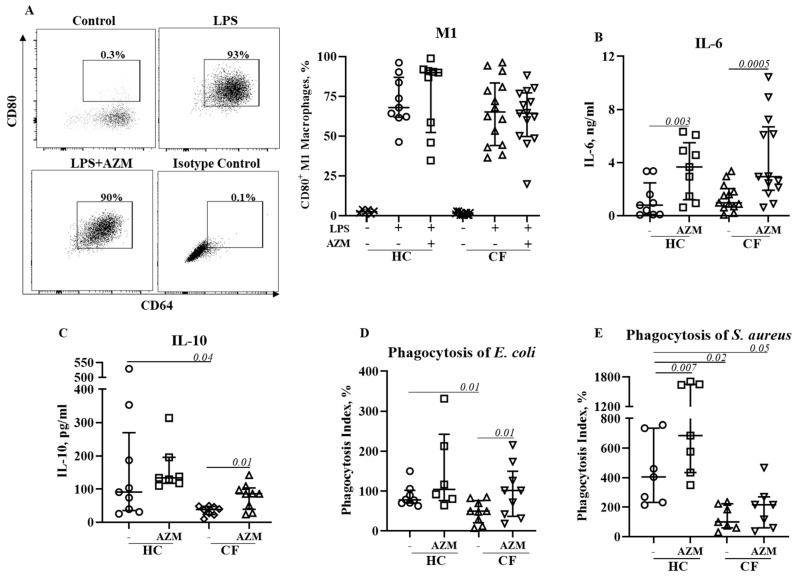
Effect of AZM on LPS stimulated M1 macrophage polarization and pro-inflammatory cytokine release. Monocytes isolated from healthy controls (HC) or from patients with CF were stimulated with GM-CSF for 6 days to differentiate them into unpolarized M0 macrophages in the presence or absence of AZM. M1 polarization was induced by *E. coli* LPS (20 ng/mL). The population frequencies of the CD80^+^ M1 macrophages were analyzed (**A**). The release of IL-6 (**B**), IL-10 (**C**) from the M1 macrophages was quantified in the supernatant. Bacterial phagocytosis was studied using pHrodo *E. coli* green (**D**) and pHrodo *S. aureus* red (**E**) bioparticles. Each symbol represents an individual healthy donor (HC = 6–9) or pwCF (N = 7–14). Wilcoxon signed-rank and Mann–Whitney tests were performed. The data are shown as the median (25%, 75%).

**Figure 4 cells-13-00166-f004:**
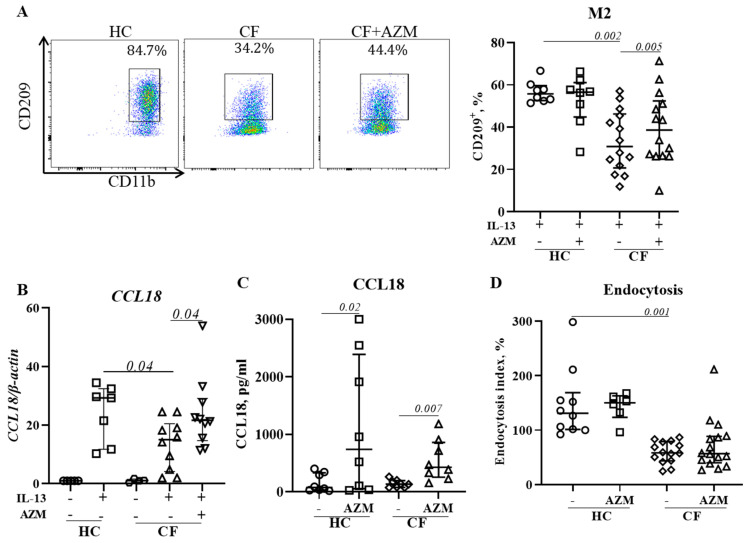
AZM promotes anti-inflammatory (M2) macrophage polarization in CF. MDMs were stimulated to M2 polarization by IL-13 (20 ng/mL) with or without AZM. Flow cytometric plots of CD209^+^ M2 macrophages from representative HCs and CFs and population frequencies of CD209^+^ M2 (**A**) were analyzed. To confirm the M2 polarization, the mRNA level of the M2-specific CCL18 gene was studied (**B**). CCL18 release was quantified in the culture supernatant (**C**). The endocytosis of polarized M2 macrophages was analyzed using AF647-labelled dextran (10 KD) (**D**). The endocytic index was calculated by normalizing the MFI with the number of CD209^+^ M2 macrophages. Each dot represents an individual donor (HC = 7–8) or pwCF (N = 8–14). Wilcoxon signed-rank and Mann–Whitney tests were performed. The data are shown as the median (25%, 75%).

**Table 1 cells-13-00166-t001:** Patient demographic and clinical characteristics of the patients with CF.

	Adults	Children
n	13	10
Age, years (range)	23–55	8–12
Gender, female	4 (30.8%)	5 (50%)
Genotype, n (%)		
Phe508del homozygous	7 (54%)	5 (50%)
Phe508del heterozygous	6 (46%)	5 (50%)
Lung function *, (mean ± SD)		
FEV1, L	1.8 ± 0.85	1.8 ± 0.36 *
FVC, L	3.5 ± 1.36	2.0 ± 0.4 *
*Pseudomonas aeruginosa* infection status, n (%)		
Chronic	12 (92.3%)	6 (50%)
Intermittent	1 (7.7%)	2 (25%)
Never	0	2 (25%)

* Data point missing for one study participant.

**Table 2 cells-13-00166-t002:** List of antibodies used for flow cytometry and western blot.

Antibody	Cat#	Supplier	Used in
TLR4 AF488	#539917	eBioscience	Flow cytometry
IL-13Rα1 APC	#360406	BioLegend	Flow cytometry
IL-4Rα PE Cy 7	#355008	BioLegend	Flow cytometry
CD80 PE	#305208	BioLegend	Flow cytometry
CD64 PE Cy7	#305022	BioLegend	Flow cytometry
CD209 BV421	#330117	BD	Flow cytometry
CD11b APC	#550019	BD	Flow cytometry
7-AAD	#559925	BD	Flow cytometry
GAPDH	#2118	CST	Western blot
Phospho-p65	#3031	CST	Western blot
Phospho-ERK1/2	#9272	CST	Western blot
Total ERK1/2	#9102	CST	Western blot

Corresponding isotype controls were used.

## Data Availability

All the data has been stored in the research data management system of The University of Queensland (RDM-UQ).

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
