# Peer review of "Azithromycin Augments Bacterial Uptake and Anti-Inflammatory Macrophage Polarization in Cystic Fibrosis"

_cells, 2024, doi:10.3390/cells13020166_

Round 1
Reviewer 1 Report
Comments and Suggestions for Authors
In this study, monocytes, isolated from the venous 18 blood of patients with cystic fibrosis (pwCF) and healthy controls (HCs) were differentiated into monocyte-derived macrophages (MDMs) and infected with P. aeruginosa. P. aeruginosa uptake and killing by MDMs in the presence or absence of azithromycin was studied. Azithromycin enhanced uptake was associated with ERK1/2 pathway upregulation. Increased phagocytosis of E. coli and S. aureus was also observed by cystic fibrosis (CF) pro-inflammatory M1 macrophages. In addition, azithromycin partially restored anti-inflammatory M2 macrophage polarization in CF. Such enhanced uptake conferred improved bacterial killing in CF and restoration of anti-inflammatory M2 polarization may contribute to the clinical improvement seen in patients with CF treated with AZM.
Technical comments:
Please include a materials section mentioning the materials purchased for conducting the study including the company name and place. For example: I didn’t find the name of the company of azithromycin. What is GM-CSF?
Minor comments:
Page #6, line #:193“was previously reported defective in CF”, is to be written as “was previously reported to be defective in CF”. Please correct the sentence.
Page #4, line #131: Please insert the sign of degree after 37, in the sentence “Cells were infected with PA14 at MOI 10 for 1hr at 37C.”.
Gram positive and negative should have the name of the Scientist Gram in capital letters throughout the manuscript.
I recommend minor revision.
Author Response
Please include a materials section mentioning the materials purchased for conducting the study including the company name and place. For example: I didn’t find the name of the company of azithromycin. What is GM-CSF?
Response: We thanked the reviewer for the comment. Method section has been updated as per reviewer’s suggestion with company name and place and highlighted in yellow. Full name of GM-CSF has been added on page 3, line 101.
Minor comments:
Page #6, line #:193“was previously reported defective in CF”, is to be written as “was previously reported to be defective in CF”. Please correct the sentence.
Response: We thanked reviewer for the comment. The line has been rewritten as per reviewer’s suggestion and highlighted in yellow.
Page #4, line #131: Please insert the sign of degree after 37, in the sentence “Cells were infected with PA14 at MOI 10 for 1hr at 37C.”.
Response: We thanked the reviewer for the comment. The sign of degree after 37 has been added as per reviewer’s suggestion and highlighted in yellow.
Gram positive and negative should have the name of the Scientist Gram in capital letters throughout the manuscript.
Response: We thanked the reviewer for the comment. The word “Gram” with capital letters has been changed throughout the manuscript. The sign of degree after 37 has been added as per reviewer’s suggestion and highlighted in yellow.
Reviewer 2 Report
Comments and Suggestions for Authors
The abstract is in an unusual form - I suggest rewording.
Literature cited in round brackets instead of square brackets.
Subsection titles once ending with a full stop, once not.
Table 2 - unclear. What is meant by WB? Table title 'used for flow cytometry, microscopy and immunoblot', only flow cytometry listed.
Author Response
The abstract is in an unusual form - I suggest rewording.
Response:
Literature cited in round brackets instead of square brackets.
Response: I thanked the reviewer for the comment. References were changed to square brackets throughout the manuscript.
Subsection titles once ending with a full stop, once not.
Response: I thanked the reviewer for pointing out this issue. A full stop has been added in all subsections.
Table 2 - unclear. What is meant by WB? Table title 'used for flow cytometry, microscopy and immunoblot', only flow cytometry listed.
Response: I thanked the reviewer for pointing this issue. Table 2 has been updated with western blot. Table title has also been updated and highlighted in yellow.